**Data Availability Statement:** Data cannot be shared publicly because of Privacy policy of the Indian Council of Medical Research, India. Data are

# Hepatitis C virus seroprevalence among patients enrolled at the opioid substitution therapy center in Bihar: A cross-sectional study

Ashish Kumar[‡], Hemant Mahajan[¤‡], Sanjay Chaturvedi, Ashok Kumar, Shiril Kumar, Ganesh Chandra Sahoo, Vidya Nand Rabi Das, Krishna Pandey*

Indian Council of Medical Research—Rajendra Memorial Research Institute of Medical Sciences, Patna, India

¤ Current address: Department of Public Health Nutrition, Indian Council of Medical Research–National Institute of Nutrition, Hyderabad, Telangana, India

‡ AK and HM are contributed equally to this work and share first authorship on this work.

* drkrishnpandey@yahoo.com

## Abstract

### Background and aim

Hepatitis C virus (HCV) infection poses a major public health challenge in Indian settings due to its huge population and easy transmissibility of HCV among individuals who inject drugs (PWID, which is increasing in India). The National AIDS Control Organization (NACO), India has started the Opioid Substitution Therapy (OST) centers to improve the health status of opioid dependent PWID and prevent the spread of HIV/AIDS among them. We conducted a cross-sectional study to find out the HCV sero-positive status and associated determinants in patients attending the OST centre in the ICMR-RMRIMS, Patna.

### Materials and methods

We utilized the routinely collected (as a part of the National AIDS Control Program) and de-identified data from the OST center from 2014 to 2022 (N = 268). We abstracted the information for exposure variables (such as socio-demographic features and drug history) and outcome variable (HCV serostatus). The association of exposure variables with HCV serostatus was examined using robust Poisson regression.

### Results

All the enrolled participants were male and the prevalence of HCV seropositivity was 28% [95% confidence interval (CI): 22.7% - 33.8%)]. There was a rising prevalence of HCV seropositivity with number of years of injection use (p-trend <0.001) and age (p-trend 0.025). Approximately, 6.3% participants were injecting drugs for >10 years and reported the maximum prevalence of HCV seropositivity (47.1%, 95% CI: 23.3%-70.8%). In adjusted analyses, being employed compared to unemployed patients [adjusted prevalence ratio (aPR) = 0.59; 95% CI: 0.38–0.89]; graduated patients compared to illiterate patients [aPR = 0.11;

available from the ICMR-RMRIMS Institutional Data Access / Ethics Committee (contact the member secretary Dr. Vidya Nand Rabi Das at drvnrdas@yahoo.com) for researchers who meet the criteria for access to confidential data.

**Funding:** The author(s) received no specific funding for this work.

**Competing interests:** The authors have declared that no competing interests exist.

95% CI: 0.02–0.78]; and patients with education up to higher secondary compared to illiterate patients [aPR = 0.64; 95% CI: 0.43–0.94] had significantly lesser HCV seropositivity. A-one year increase in injection use [aPR = 1.07; 95% CI: 1.04–1.10] was associated with 7% higher prevalence of HCV seropositivity.

## Conclusions

In this OST center-based study of 268 PWIDs residing in Patna, ~28% of patients were HCV seropositive, which was positively associated with years of injection use, unemployment, and illiteracy. Our findings suggest that OST centers offer an opportunity to reach a high-risk difficult to reach group for HCV infection and thus support the notion of integrating HCV care into the OST or de-addiction centres.

## Introduction

Despite a low prevalence in general population, hepatitis C virus infection (HCV) poses a major public health challenge in Indian setting due to its large population; India contributes a huge share of the global burden of HCV. The general population of India has HCV sero-prevalence of 0.4% to 1.0% [1], which is higher than many south east Asian countries (such as Bhutan, 0.47%, DPR Korea, 0.34%; Maldives, 0.41%; Nepal, 0.38%; Sri-Lanka, 0.23%; South East Asia Region, 0.84%) [2]. Thus at a given point in time, ~5.5 million patients are HCV seropositive and over the period of next 20 to 30 years ~1.5 million of these will develop chronic hepatic illnesses including cirrhosis and hepatocellular carcinoma if not treated promptly and appropriately [3–5].

Health care related procedures, injection drug use, and blood transfusions contribute to the spread of the HCV [6]. Since the beginning of blood product screening before blood transfusion, HCV transmission through blood transfusion has been significantly reduced and currently the injection drug use remains as the main mode of transmission of HCV in developed as well as many developing countries like India. Among persons who inject drugs (PWID), HCV transmitted either by direct or indirect blood-to-blood contact as well as using injection equipment shared directly or indirectly [6]. Thus, PWID are mainly driving the HCV epidemic in many countries. Moreover, HCV infection commonly occur in PWID when they are young; such patients are at high risk to develop chronic HCV and could face huge health care expenses [7]. If not treated appropriately, they could also transmit HCV to other individuals. The healthcare expenses of caring for people with HCV could further compromise an already fragile health care system in country like India [8].

Opioid substitution therapy (OST) centers have been developed over the course of the last three decades to treat individuals with opioid use disorder (OUD) and have now escalated at national level as a part of National AIDS Control Organization, India (NACO) to improve the health status of PWID with OUD and prevent spread of HIV/AIDS among them [9]. Traditionally, it has been seen that ~15% PWID has HIV/AIDS and ~50% of the PWID have HCV [9]. Thus, OST centers provide a good opportunity to screen these patients for HCV, understand its epidemiology and natural history and connect them to comprehensive HCV care.

Bihar, one of the eastern states in India, has the least prevalence of OUD (0.1%) compared to the national average of 0.7% [10, 11] However, anecdotal evidence suggests the recent rise in opioid users (which is partly attributable to alcohol prohibition since 2016). The news

reports are repeatedly reporting that the de-addiction centres in Bihar have been receiving an increased number of cases of substance abuse, ranging from cannabis, inhalants, and sedatives to opioids. This claim is further supported by the specific activities of the state government such as capacity building of Medical Officers of the state by the National Institute of Mental Health and Neurosciences, Bengaluru to manage the sudden increase in the number of drug abuse cases [12, 13]. Generally, opioid users initially get addicted to oral opioids and later on switch to injectable, which carries a high risk of HCV infection [8]. Thus, the recent shift in opioid consumption in Bihar could fuel the epidemic of HCV over a period of time and calls for urgent public health attention [8]. In Bihar, several medical colleges and NGOs are having OST centres, which conduct routine testing for HCV serostatus. However, no report so far has been published, which accurately describes the distribution and determinants of HCV among these OST attendees. To plan an efficient public health response to curb the epidemic of HCV (especially among PWID who are the main driver of HCV epidemic), it is important to have reasonably accurate epidemiological data.

Therefore, we have conducted a study to find out the sero-prevalence of anti-HCV antibody in patients attending an OST centre in ICMR-RMRIMS, Patna and to study the determinants (such as socio-demographic and behavioural factors) of HCV sero-positivity. The government of Bihar, as a part of National Viral Hepatitis Control Program, India (launched in 2018), is planning to distribute free antivirals (directly acting agents) to HCV patients through medical colleges and other designated institutes such as ICMR-RMRIMS, Patna from the year 2022. The findings of the study may further inform the government's initiative to integrate HCV care into OST center.

## Materials and methods

### Study population

The OST clinic was started at the ICMR-RMRIMS, Patna in January 2014 as a part of NACO's mission (funded by Ministry of Health & Family Welfare, India) to reduce the transmission of HIV among PWID. Whenever a patient visits OST center (to seek services), the OST medical officer conducts a structured interview and examination of the patient to assess demographic, behavioral, and clinical history including drug use and opioid dependency and enroll him/her based on the OST suitability criteria: (a) Opioid dependence with a long history; (b) Current use of injection drug; (c) Age ≥18 years; (d) No medical contraindication to OST; (e) Ability to give informed consent; (f) Failed attempts at achieving abstinence through other means (detoxification); (g) Feasibility to comply with requirements of OST center (such as staying close by and ability to visit the center on daily basis). Once enrolled in OST center, the trained medical officer starts buprenorphine at the clinic. The OST patients are supposed to visit the center on daily basis to receive the directly observed buprenorphine drug therapy. All patients at baseline also undergo testing to detect HIV, HBV antigen, and HCV antibody, which are routinely done at the ICMR-RMRIMS, Patna. If found HIV positive, the medical officer links patients to the designated ART centers to receive appropriate HIV management. However, for HBV antigen positive and HCV seropositive patients, the medical officer only counsels patients and refer to tertiary healthcare center for further management.

The study complied with the principles of Helsinki Declaration and obtained ethics approvals from the ICMR-RMRIMS, Patna ethics committee. As we were using the archived data collected as a part of routine national program activity, the ethics committee of the ICMR-RMRIMS, Patna waived the requirement for informed consent from the participants (Approval letter no. RMRI/EC/34/2022; Dated: 03 August, 2022). For this study, we included OST patients who were enrolled and had undergone HCV serostatus assessment (at the time

of enrollment) from January 2014 to December 2022 at the ICMR-RMRIMS. During this time, total 268 patients were enrolled and had undergone the HCV serostatus assessment.

## Variables and data sources

This study utilized routinely collected and de-identified data from the OST center at the ICMR-RMRIMS, Patna. Authors have abstracted information for various exposures and outcome variable from the medical records of OST patients.

## Outcome variable

'HCV sero-status' was the primary outcome of the study. The 'HCV serostatus' is being routinely assessed at the ICMR-RMRIMS using in-vitro qualitative enzyme linked immunosorbent assay (J. Mitra and Co. Ltd., New Delhi, India) for the detection of antibodies against HCV (anti-HCVs) in human serum and considered as HCV sero-positive if they have HCV antibody.

## Exposure variables

The extracted information on exposure variables include: 'Socio-demographic variables' such as age in years (completed), age of injection drug onset in years, sex (male; female), education level (illiterate, upto higher secondary, and more than higher secondary), marital status (unmarried, married and separated or divorced), and occupation (student, employed, unemployed, part-time and full time employment); 'Drug history' included type of drugs used (such as buprenorphine, pentazocin, heroin, cannabis, chlorpheniramine etc.), type of drug administration, and frequency of drug administration in years, and history of injection equipment sharing.

## Statistical analyses

We have used mean, standard deviation, frequencies, percentage, medians, and interquartile ranges (IQR) to describe the distribution of variables in the participants. Across HCV status (absent vs. present): (a) Age was expressed as means and standard deviation (SD) and compared using unpaired t-test; (b) Number of injection years and total number of drug used as medians and interquartile range (IQR) and compared using Mann-Whitney-U test; and (c) Categorical variables such as marital status, education, occupation, type of drug use, HIV-, HAV-, HBV-, HEV-status were expressed in percentages and compared using Pearson's chi-squared test.

We also examined the prevalence of HCV across the categories of age (18 to 20, 21 to 25, 26 to 30, 31 to 35, 36 to 40, and >40 years) and categories of number of years of injection use (≤2, 2.1 to 5, 5.1 to 10, and >10 years) and assessed the significance using linear trend test. We used Poisson regression with a robust variance estimator [14] to calculate unadjusted and adjusted prevalence ratio (PR) for various socio-demographic variables and clinical parameters (with HCV status as an outcome). All p-values were two-tailed and a p-value <0.05 was considered as significant. Stata version 14.2 (Stata Corp, College Station, TX, USA) was used for all statistical analyses.

## Results

From January 2014 to December 2022, total 268 opioid dependent PWID were enrolled at the OST center. The mean (SD) age of study participants was 31.59 years (9.03); ~85.0% participants were between ages 18 to 40 years. The age ranged from 18 years to 59 years. All the study participants were males and ~30% were unmarried. ~22% participants were illiterate and

**Table 1. Association of HCV status with various socio-demographic and clinical parameters.**

| Variables | | Total (n = 268) | HCV status | | p-value |
|---|---|---|---|---|---|
| | | | No (n = 193) | Yes (n = 75) | |
| Age in years[a], mean (SD) | - | 31.59 (9.03) | 30.85 (9.18) | 33.48 (8.41) | 0.033 |
| Marital status[b], n(%) | Unmarried | 82 (30.6) | 59 (30.6) | 23 (30.7) | 0.280 |
| | Married | 176 (65.7) | 129 (66.8) | 47 (62.7) | |
| | Separated/Divorced | 10 (3.7) | 5 (2.6) | 5 (6.7) | |
| Occupation[b], n(%) | Unemployed | 44 (16.5) | 28 (14.6) | 16 (21.3) | 0.207 |
| | Self-employed/student | 16 (6.0) | 9 (4.7) | 7 (9.3) | |
| | Part time | 49 (18.3) | 38 (19.8) | 11 (14.7) | |
| | Full time | 158 (59.2) | 117 (60.9) | 41 (54.7) | |
| Education[b], n(%) | Illiterate | 60 (22.4) | 35 (18.1) | 25 (33.3) | 0.002 |
| | Higher secondary and below | 184 (68.7) | 135 (69.9) | 49 (65.3) | |
| | Graduation and above | 24 (9.0) | 23 (11.9) | 1 (1.3) | |
| No of injection years[c], median (IQR) | - | 5 (1, 7) | 3 (1, 6) | 6 (3, 10) | <0.001 |
| Drug use[b], n(%) | Ketamine | 2 (0.9) | 1 (0.5) | 1 (1.3) | 0.486 |
| | Buprenorphine | 228 (86.4) | 162 (83.9) | 66 (88.0) | 0.402 |
| | Chlorpheniramine | 84 (30.1) | 55 (28.5) | 29 (38.7) | 0.107 |
| | Pentazocin | 63 (23.5) | 37 (19.2) | 26 (34.7) | 0.007 |
| | Heroin (Smack) | 177 (66.0) | 135 (69.9) | 42 (56.0) | 0.030 |
| | Cannabis | 117 (43.7) | 84 (43.5) | 33 (44.0) | 0.944 |
| Total number of drugs use[c], median (IQR) | - | 3 (2, 3) | 2 (2, 3) | 3 (2, 4) | 0.940 |
| HIV status[b], n(%) | Yes | 19 (7.1) | 11 (5.7) | 8 (10.7) | 0.155 |
| HAV status[b], n(%) | Yes | 6 (2.2) | 4 (2.1) | 2 (2.7) | 0.768 |
| HBV status[b], n(%) | Yes | 10 (3.7) | 6 (3.1) | 4 (5.3) | 0.388 |
| HEV status[b], n(%) | Yes | 25 (9.3) | 11 (5.7) | 14 (18.7) | 0.001 |

IQR, inter-quartile range; HIV, human immunodeficiency virus; HBV, hepatitis B virus; HCV, hepatitis C virus; n, frequency; %, percent; SD, standard deviation

[a]Age was expressed as mean (standard deviation) and compared across HCV status using unpaired t-test.

[b]Marital status, occupation, education, drug use, HIV-, HAV-, HBV-, and HEV-status were expressed as numbers (%) and compared across HCV status using Pearson's chi-square test

[c]No of injection years and total number of drugs use were expressed as median (interquartile range) and compared across HCV status using the Wilcoxon-Mann-Whitney test

~15% participants were unemployed. The HCV seropositivity prevalence was 28% [95% CI: 22.7% - 33.8%]. The OST attendees were consuming different types of opioids including buprenorphine (86.4%), pentazocin (23.5%), impure form of heroin (also known as smack, 66.0%), cannabis (43.7%), and antihistamine such as chlorpherniramine (30.1%). Approximately one quarter of the participants were consuming more than three drugs. Approximately 7% participants were HIV positive and 9.3% were HEV positive (Table 1).

Participants with HCV seropositive status (compared to participants with HCV seronegative status) were older (p-value = 0.033), illiterate or less educated (p-value 0.002), using injections for longer duration (p-value < 0.001); more likely to use pentazocin (p-value = 0.041), and less likely to use heroin (smack) (p-value = 0.030). Participants with HCV seropositivity also had higher burden of HEV compared to HCV seronegative status (p-value = 0.001). This study identified no difference in HCV status by marital status, occupation, total number of drug use, HIV-, HAV-, or HBV-status (Table 1).

The association of HCV sero-positivity and age is described in Fig 1. Prevalence was lower in the two youngest age groups compared to the four older ones, which were quite similar (p-trend = 0.025). Patients aged 25 years and less had the HCV sero-positivity of 15.7% (95% CI:

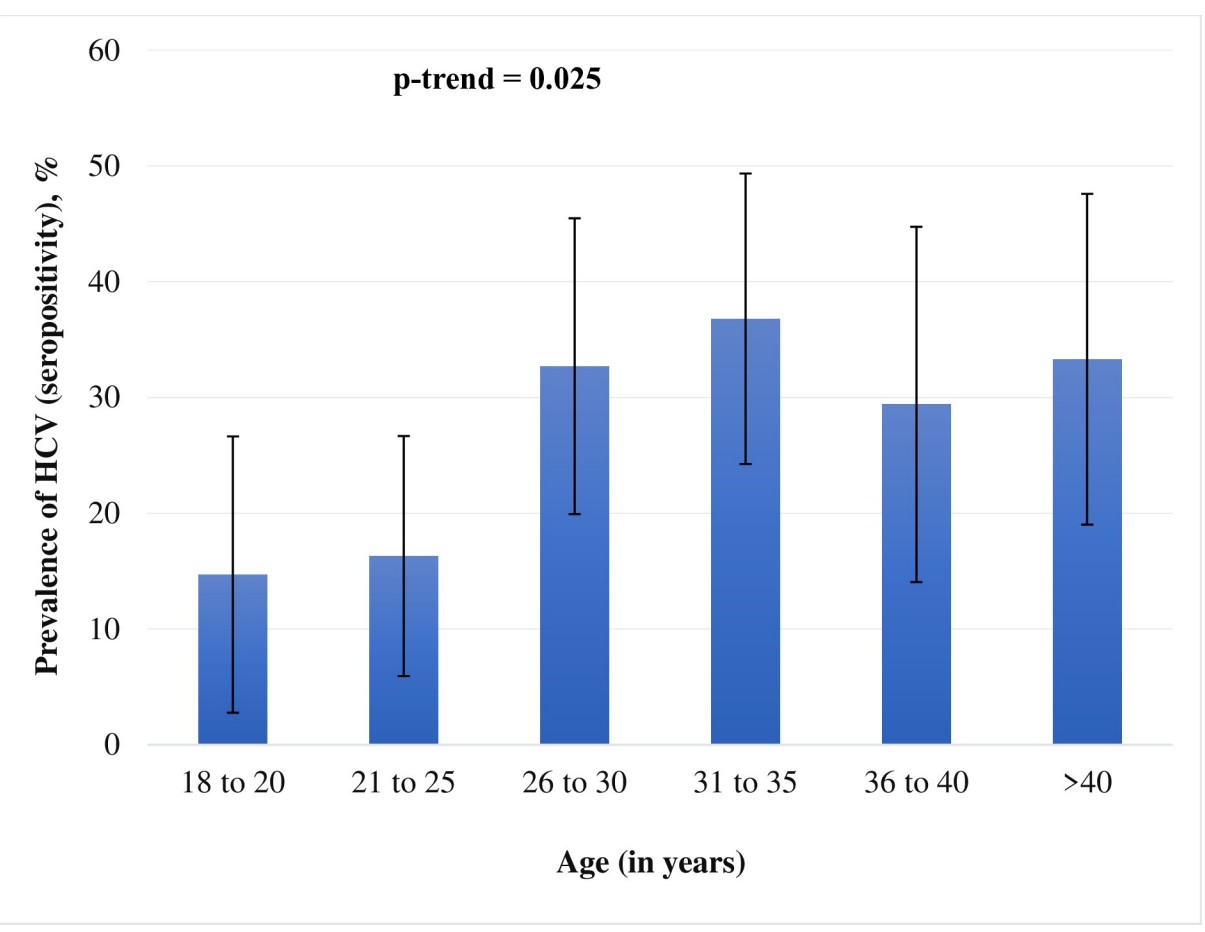

**Fig 1. Prevalence (95% confidence interval) of HCV seropositivity with age.** %, percent; HCV, hepatitis C virus infection; CI, confidence interval; Proportion of participants in each 'age group' category: 18 to 20 years– 12.7%; 21 to 25 years– 18.3%; 26 to 30 years– 19.4%; 31 to 35 years– 21.3%; 36 to 40 years– 12.7%; >40 years– 15.7%.

8.6% - 25.3%); whereas patients older than 25 years of age had the HCV sero-positivity of 33.5% (95%CI: 26.8% - 40.8%).

The association of HCV sero-positivity and years of injection drug use is described in Fig 2. Prevalence was higher in each group with more years of injecting compared to the group with fewer years—a monotonic increase in prevalence (p-trend <0.001). The prevalence of HCV seropositivity was 15.6% (95% CI: 8.3% - 22.9%), 28.4% (95% CI: 18.1% - 38.7%), 38.3% (95% CI: 27.7% - 48.9%), and 47.1%, 95% CI: 23.3%-70.8% among participants who were injecting drugs for ≤ 2 years, >2 to ≤5 years, >5 to ≤10 years, and >10 years, respectively.

Among several sociodemographic and clinical variables, occupation, education, and years of injection use were significantly associated with HCV sero-positivity (Table 2). In a multivariable robust Poisson regression, employed compared to unemployed patients were 41% [adjusted PR (aPR) (95% CI): 0.59 (0.38, 0.89)] less likely to have HCV positivity (p-value <0.05). Patients with education up to higher secondary compared to illiterate patients were 36% [aPR (95% CI): 0.64 (0.43, 0.94)] less likely to have HCV sero-positivity (p-value <0.05). Whereas, patients with graduation and above compared to illiterate patients were 89% [aPR (95% CI): 0.11 (0.02, 0.78)] less likely to have HCV sero-positivity (p-value <0.05). A-one year increase in injection use was associated with 7% [aPR (95%): 1.07 (1.04, 1.10)] higher prevalence of HCV sero-positivity (p-value < 0.05).

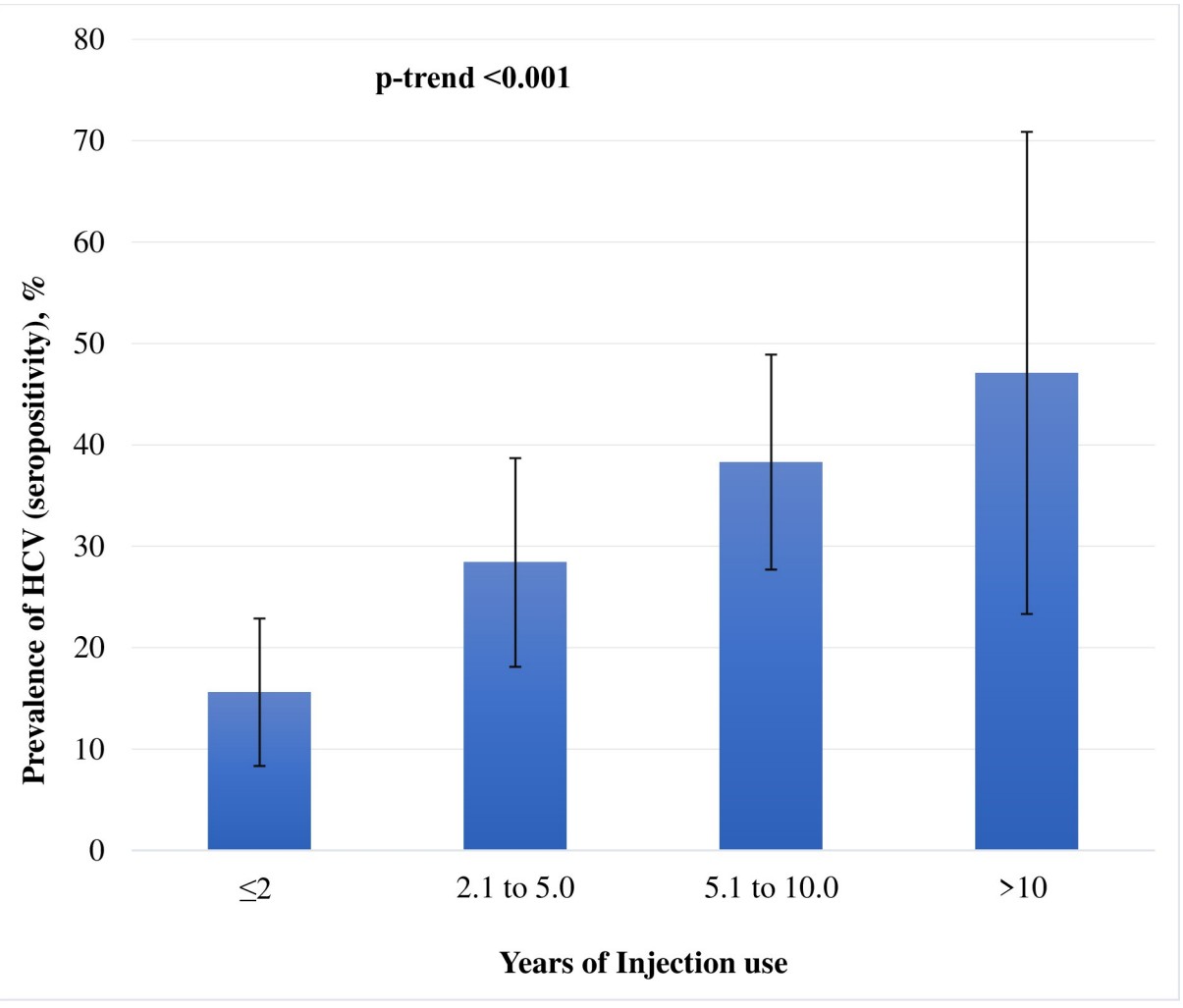

**Fig 2. Prevalence (95% confidence interval) of HCV seropositivity with number of years of injection use.** %, percent; HCV, hepatitis C virus infection; CI, confidence interval; Proportion of participants in each category of 'years of injection use': ≤2 years of injection use– 35.8%; 2.1 to 5 years of injection use– 27.6%; 5.1 to 10 years of injection use– 30.2%; >10 years of injection use– 6.3%.

## Discussion

Among 268 patients enrolled in OST center at ICMR-RMRIMS, Patna from 2014 to 2022, we assessed the prevalence and predictors of HCV serostatus. All the enrolled participants were male. This is the first study to our knowledge that assessed the prevalence and predictors of HCV serostatus among OST patients in Bihar. Our data suggest a high prevalence of HCV seropositivity among OST patients. We also found that those unemployed, less educated, and who were using injection drugs for longer period of time had a higher HCV seropositivity than their counterparts. The observed findings have significant public health significance to understand the burden of HCV among PWID enrolled in OST center and call for an urgent need to prevention and treatment strategies in this high risk group.

### Burden of HCV sero-positivity in OST center attendees

The high prevalence of HCV among OST attendees in ICMR-RMRIMS is consistent with other regions of India [1]. The observed HCV prevalence of 28% among OST patients in Bihar

**Table 2. Determinants of HCV seropositive status among patients enrolled at the OST center, RMRIMS, Patna: Prevalence ratio (95% confidence interval).**

| Variables | Categories | Unadjusted | Adjusted |
|---|---|---|---|
| | | PR (95% CI) | PR (95% CI) |
| Age (in years) | Per year increase | 1.02 (1.01, 1.04) | 1.00 (0.98, 1.03) |
| Marital status | Unmarried | 1.00 | 1.00 |
| | Married | 0.95 (0.62, 1.46) | 0.88 (0.57, 1.35) |
| | Separated/divorced | 1.78 (0.88, 3.63) | 1.47 (0.71, 3.05) |
| Education | Illiterate | 1.0 | 1.0 |
| | Higher secondary and below | 0.64 (0.43, 0.94) | 0.64 (0.43, 0.94) |
| | Graduation and above | 0.10 (0.02, 0.70) | 0.11 (0.02, 0.78) |
| Occupation | Unemployed | 1.00 | 1.0 |
| | Employed | 0.72 (0.46, 1.13) | 0.59* (0.38, 0.89) |
| Year of injection use | Per year increase | 1.06 (1.04, 1.09) | 1.07** (1.04, 1.10) |
| No. of drug use | Per drug use increase | 1.14 (0.94, 1.40) | 1.09 (0.92, 1.28) |

CI, confidence interval; PR, Prevalence ratio

For the multivariable robust Poisson regression model (adjusted analysis), we included variables: age, marital status, education, occupation, number of years of injection use and total number of drugs used.

*PR of 0.59 means that employed participants were 41% less likely to have HCV sero-positive status compared to unemployed participants after adjusting for the effect of potential confounders on the HCV serostatus.

**PR of 1.07 means that a-one year increase in year of injection use made a participant in this study 7% more likely to have the HCV sero-positive status after adjusting for the effect of potential confounders on the HCV serostatus.

was significantly less than the 44.7% prevalence reported by Goel et al. in their systematic analysis of HCV in PWID across India [1]. This finding may reflect differences between the OST patients and community-based sample of PWID with regard to education attainment, social support, behavioural risks, or years of injection drug use etc. The differences in HCV sero-prevalence could also be attributed to geographical disparities in terms of access to injectable opioids [11]. Nevertheless, our findings suggest that approximately 30% of patients enrolled in OST center was HCV seropositive, which is worrisome and demands urgent attention from policymakers. The observed HCV sero-prevalence is ~33 times higher than the HCV sero-prevalence reported in the general population of India i.e. 0.85% (95%CI: 0.00%-3.98%) [1, 2]. Several clinical, social and behavioural factors may drive the higher prevalence of HCV among PWID. The level of exposure to adverse risk environments is found to be higher in PWID compared to others [7]. Compared with the general population, PWID are at greater risk of police arrest, incarceration, sex work, and the experience of homelessness or unstable housing, all of which are associated with increased blood-borne virus transmission [7]. Unfortunately, the available clinical records for the OST attendees in ICMR-RMRIMS do not provide any information about the mentioned adverse exposures, prohibiting comparison with other studies [7].

## Determinants of HCV sero-positivity and comparison with other studies

The determinants of HCV sero-positivity (such as education, employment and years of injection drug use) observed in this study are very similar to those reported in other studies among PWID. Consistent with the previously reported literature, we found the robust positive association between years of injection use and high HCV sero-positivity [15–18]. Similarly, HCV sero-positivity was lower among educated- and employed-patients [15–18]. In this study, 31% (95% CI: 25.5%– 36.9%) PWID were age less than or equal to 25 years (considered as 'young

PWID'). This is consistent with the estimates reported by Degenhardt et al. 27.8% (95% CI: 19.0%– 36.7%) for South Asia [7]. We also observed a sharp increase in the HCV sero-positive status after the age of 25 years. There was a positive linear trend for HCV positive status with increase in age of participants.

## Public health significance of the findings

The findings of the study may further support the Bihar government's initiative to integrate HCV care into OST center for several reasons. First, an OST center provides an ideal opportunity to screen high risk group for HCV. Second, the high prevalence of HCV seropositivity among OST center attendees means easy and early access to many HCV patients (which generally are hard to reach population) and initiation of prompt treatment to reduce a reservoir pool of HCV. Third, the directly observed buprenorphine therapy (at the OST center) stabilizes PWID (as it provides slower onset but longer duration of action) [9], reduces their cravings for injections and improves patients' compliance with HCV therapy and thus, may prevent further HCV transmission in a community. Fourth, an OST center provides an enabling environment to provide HCV treatment along with risk-reducing behaviour measures–a well-established way to treat HCV and prevent re-infection as well as HIV transmission [9]. Fifth, OST center requires patients to visit either daily or at short intervals (to receive directly observed buprenorphine therapy) thus may provide an ideal setting to monitor HCV care across continuum from point-of-care diagnosis to treatment and follow-up care and reinfection prevention counselling. The OST center set-up has been successfully utilized to provide HIV management in multiple national and international settings [9, 19]. The approaches used and lessons learned from these settings could be utilized in Indian context to efficiently integrate the HCV care in existing OST center set-up to reduce occurrence of new cases as well as treatment of prevalent cases.

## Strengths and limitations

The strength of the study include: (a) Complete data on HCV sero-status for all enrolled patients; and (b) Standardized sero-status assessment of patients at the ICMR-RMRIMS, Patna. There are several limitations to this study that should be taken into account. First, a very small sample size; The OST center had enrolled, on average, fewer than 30 patients every year. The barriers to entry were—active injection use, daily or near daily visits to the center, failure of non-medication treatment for oral use of drugs—made for not only a small but potentially very biased sample. Second, limited external validity of findings; the study included only opioid dependent PWID, who were seeking services from the OST center. Moreover, all the enrolled participants were males. Therefore, we believe that the study participants may not represent the true population of PWID in the community. Third, the medical records at the OST center in ICMR-RMRIMS, Patna has information for limited number of determinants (mostly limited to socio-demographic and years of injection use). As per the NACO guidelines for OST centers in India [11], the OST centers are supposed to collect complete information on socio-demographics, complete drug use history, clinical history, mental health, psychosocial health, adverse exposures etc. However, the medical records of the study participants had no information about behavioural risk factors (such as smoking, alcohol etc.) as well as adverse exposures such as history of police arrest, incarceration, sex work, sharing of injection equipment or experience of homelessness or unstable housing. Nevertheless, the primary objective of the study was to understand the burden of HCV seropositive status; and all the study participants had undergone through a standardized process to assess their HCV antibody status. Fourth, OST center has information for HCV antibody (an evidence of past infection that

could have become chronic) and not the viral load which is a better indicator of current HCV infection. Fifth, the availability of directly acting agents through OST center in Patna (since 2022) could have increased the enrolment of HCV patients at the OST center (potentially inflating the HCV seropositive burden among OST attendees). However, this is very unlikely in our study as only one seropositive patient was enrolled at the OST center in the year 2022.

## Conclusion

In this OST center-based study of 268 PWID residing in Patna, Bihar, 28% of participants were HCV seropositive. The HCV seropositive status was positively associated with an increased years of injection use, unemployment, and illiteracy. Our findings support the notion of integrating HCV care into OST center or de-addiction centre where comprehensive HCV care could be delivered on regular basis. Furthermore, the feasibility and effectiveness of HCV care through OST clinic or de-addiction centre needs to be examined using controlled trials to identify optimal strategy of HCV-care integration.

## Author Contributions

**Conceptualization:** Ashish Kumar, Hemant Mahajan.

**Data curation:** Ashish Kumar, Hemant Mahajan.

**Formal analysis:** Hemant Mahajan.

**Methodology:** Ashish Kumar, Hemant Mahajan.

**Supervision:** Ashok Kumar, Krishna Pandey.

**Writing – original draft:** Hemant Mahajan.

**Writing – review & editing:** Ashish Kumar, Sanjay Chaturvedi, Ashok Kumar, Shiril Kumar, Ganesh Chandra Sahoo, Vidya Nand Rabi Das, Krishna Pandey.

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
