## [Decision Letter · Decision Letter 0]

26 Jan 2023

PONE-D-22-32802The Burden of HCV among Patients Enrolled at the Opioid Substitution Therapy Clinic in Bihar: A Cross-Sectional StudyPLOS ONE

Dear Dr. Pandey,

Thank you for submitting your manuscript to PLOS ONE. After careful consideration, we feel that it has merit but does not fully meet PLOS ONE’s publication criteria as it currently stands. Therefore, we invite you to submit a revised version of the manuscript that addresses the points raised during the review process.

We look forward to receiving your revised manuscript.

Kind regards,

Nickolas Zaller

Academic Editor

PLOS ONE

Journal Requirements:

4. Please amend your authorship list in your manuscript file to include author Vidya Nand Rabi Das.

Reviewers' comments:

Reviewer's Responses to Questions

**Comments to the Author**

1. Is the manuscript technically sound, and do the data support the conclusions?

Reviewer #1: Partly

Reviewer #2: Partly

2. Has the statistical analysis been performed appropriately and rigorously? 

Reviewer #1: Yes

Reviewer #2: Yes

3. Have the authors made all data underlying the findings in their manuscript fully available?

Reviewer #1: No

Reviewer #2: No

4. Is the manuscript presented in an intelligible fashion and written in standard English?

Reviewer #1: Yes

Reviewer #2: No

5. Review Comments to the Author

Reviewer #1: Summary

This study used medical records from an OST clinic to estimate the HCV prevalence among clients and identify characteristics associated with HCV. The authors use the findings to reiterate their support for increased resources to integrate HCV testing into OST clinics in Bihar specifically, and India generally.

Overall the methods are sound but several steps need clarification. The discussion stretches the beyond the scope of the findings quite a bit. The authors attempt to contextualize their findings within a larger narrative of injection drug use, but offer quite a bit of personal opinion on the matter and use stigmatizing and negative language that is not appropriate.

General comments

The use of the terms ‘addicts’ and ‘addicted’ are outdated and not appropriate the journal’s audience. Individuals with an opioid use disorder is one alternative.

“Smack” is not an appropriate term for heroin. Although this is a common street name, readers may interpret it differently.

There are substantial typos and grammatical errors throughout; this paper cannot be typeset and published as it is.

Highlights

“Very high” may be misleading, as this study identified a lower HCV prevalence than was identified in the India-based meta-analysis the authors cited.

“Effective and evidence-based care can be delivered…” is beyond the scope of this study’s findings. While this is the final recommendation the authors provide, this statement as a highlight should be tempered to align with the study findings.

Introduction

Lakh should probably be written as thousand for non-Indian readers.

There is a claim that increasing opioid use is attributable to alcohol prohibition. This needs further explanation and a citation.

The government is planning to distribute DAA starting in 2022. Is there more recent data available? Is there any information in the dataset about DAA receipt? This would definitely be a confounder and if not available should be included as a limitation.

Methods

Justify the time frame selected for the medical record extraction. Were previous years or more recent years available, and if so, why were 2014-2020 selected?

Clarification on the final sample size is needed. Who was excluded? 25% were lost to follow up – how does this relate to the final analytic sample of n=213?

The authors state that they did not adjust for drug type because there is no literature supporting its inclusion. The authors didn’t include justification from the literature to include the other variables, so it seems inconsistent. To that end, there certainly is literature around HCV and treatment (buprenorphine) and drug use (heroin and cannabis) which are drug types included in the data. In addition, alcohol has is strongly associated with HCV but is not included. Further explanation and justification for these analytic variable decisions is needed.

Results

The authors make several statements that are not supported by their findings, specifically regarding p-values. For example, stating that clients with HCV were older, but this is not supported by the proportions in Table 1 or by the p-values. This study identified no difference in HCV status by age.

There needs to be a full description of all the variables included in the study and why. For example, Avil and Penta were not mentioned in the methods section but showed up in the results tables. Why are antihistamines included in the analysis?

The results include that a higher number of deaths were observed among HCV clients. Death is not defined in the methods section. How was death defined? Death for any reason during the study period? In this case, death would be conceptualized as an outcome, as HCV infection would necessarily precede mortality. Either explain this sub-analysis and the justification for it better, or remove it.

The patient status variable in Table 1 is confusing and was not described in the methods section. The term “on drugs” is ambiguous. Also, the inclusion of those who were lost to follow or migrated is difficult to follow. As a cross-sectional study there was no follow up. Are these categories mutually exclusive? Regardless of how you define ‘on drugs’, can a client be on drugs and migrate to another facility? This variable needs to be described in detail in the methods.

The resolution of the figure images is very poor and not fit for publication. Provide these to the journal as vector graphics.

The trend for age is not significant, but stated using language that suggests a trend was observed. The same for years of injection use. It is also unclear which model these trends were derived from, which should be clarified in the methods.

Injection was significant in the full model, but it is not stated clearly where the estimates for the figures were derived from.

The decision to conduct two separate models – restricted and full – needs more rationale and justification as to what one model tells us that the other cannot. But, it seems as if the authors only present the results from the full model anyway, so the purpose of the restricted is unknown.

The variable selection criteria of p<0.10 from the univariable models needs a citation.

Discussion

Burden paragraph

The authors state that patients in OST don’t represent true population of IDUs – why? This needs to be explained further and supported with citations.

The authors state that OST and non-OST populations differ because the former is more responsible. This is not supported in the literature and is highly judgmental and stigmatizing and language that perpetuates clinical treatment disparities. This is an opinion and not appropriate for publication.

The authors state that they can confidently say they underestimated HCV – why? How? Overestimating is also plausible, in that people with more severe disorders and higher HCV risk are the ones who get treatment. How the authors are conceptualizing underestimation, and how they use the epidemiologic and drug use literature to back it up, are needed.

The statement “IDU live alone in poverty”, and several other similar statements in this paragraph, is not supported by the literature, which is likely the reason the authors offer no citations. Living alone and poverty are certainly determinants of drug use, but these characteristics do not universally describe the IDU population.

History of crime is also not cited – again, this is stigmatizing language and projects a judgement from the authors. Poverty, exposure to the criminal justice system, and the resulting stress all drive drug use. The relationship between crime and drug is complex and requires considerable nuance, tact, and literature review to discuss appropriately

Determinants paragraph

The claim that higher HCV indicative of worsening HCV burden is misleading and not supported by the study findings. The study found a prevalence lower than the 44.7% cited by the authors (Goel et al.). In addition, as stated earlier, the time trend was not significant. A disproportionately high and growing HCV burden is the crux of the paper and in the title, but doesn’t align with the study findings.

Public health paragraph

The authors suggest that OST is an ideal opportunity to screen for HCV. This is a great point and much of the discussion would be substantially strengthened if this were described in greater detail. People who use and inject drugs are notoriously hard to reach in communities, and there are few places where effective screening may reach a large portion of the population. OSTs are one of the few places that provide that opportunity.

Reviewer #2: Comments to Authors uploaded as a pdf file.

The formatting of my comments is complicated and the text box does not alway all of the formatting to be visible. Therefore, I have chosen to attach a pdf file containing all the comments and suggested edits.

6. PLOS authors have the option to publish the peer review history of their article (what does this mean?). If published, this will include your full peer review and any attached files.

Reviewer #1: **Yes: **George Pro

Reviewer #2: No

---

## [Author Response · Author response to Decision Letter 0]

20 Mar 2023

RESPONSE To EDITS REQUESTED

1. Please amend your authorship list in your manuscript file to include author Vidya Nand Rabi Das.

RESPONSE: We already have included author Vidya Nand Rabi Das in the revised manuscript and highlighted with yellow ink.

The revised list of Author include: Ashish Kumar, Hemant Mahajan, Sanjay Chaturvedi, Ashok Kumar, Shiril Kumar, Ganesh Chandra Sahoo, Vidya Nand Rabi Das, and Krishna Pandey;

2. Thank you for providing a Data Availability Statement and explaining why the data are restricted. Could you please provide the email address for the ICMR-RMRIMS Institutional Data Access / Ethics Committee, to which data access queries may be sent?

Response: The Member secretary of the ethics committee ICMR-RMRIMS, Patna, India is Dr. Vidya Nand Rabi Das. His Email ID: drvnrdas@yahoo.com

RESPONSE to REVIEWER COMMENTS

Response to Editor’s and Reviewers’ comments:

Editor’s comments:

• A rebuttal letter that responds to each point raised by the academic editor and reviewer(s). You should upload this letter as a separate file labelled 'Response to Reviewers'

Response: Appropriate rebuttal letter named 'Response to Reviewers' has been drafted and submitted at the designated PlosOne submission portal.

• A marked-up copy of your manuscript that highlights changes made to the original version. You should upload this as a separate file labelled 'Revised Manuscript with Track Changes'.

Response: A revised document with all the suggested modifications highlighted in Yellow has been drafted, labelled as 'Revised Manuscript with Track Changes, and uploaded at the PlosOne submission portal.

• An unmarked version of your revised paper without tracked changes. You should upload this as a separate file labelled 'Manuscript'.

Response: A revised document with all the suggested modifications has been drafted, labelled as ‘Manuscript’, and uploaded at the PlosOne submission portal.

Journal Requirements:

Response: The revised manuscript has been formatted as per the PlosOne’s style requirement.

• Please provide additional details regarding participant consent. In the ethics statement in the Methods and online submission information, please ensure that you have specified (1) whether consent was informed and (2) what type you obtained (for instance, written or verbal, and if verbal, how it was documented and witnessed). If your study included minors, state whether you obtained consent from parents or guardians. If the need for consent was waived by the ethics committee, please include this information. If you are reporting a retrospective study of medical records or archived samples, please ensure that you have discussed whether all data were fully anonymized before you accessed them and/or whether the IRB or ethics committee waived the requirement for informed consent. If patients provided informed written consent to have data from their medical records used in research, please include this information.

Response: In this study, we have used completely anonymised data from medical records of the Opioid Substitution Therapy attendees in the ‘Indian Council of Medical Research- Rajendra Memorial Research Institute of Medical Sciences, Patna, India’. We had revealed the complete information of the study to the concerned ethics committee. The need for consent was waived by the ethics committee of the ‘Indian Council of Medical Research- Rajendra Memorial Research Institute of Medical Sciences, Patna, India’ (Letter No. RMRI/EC/34/2022; Dated: 03 August, 2022). 

• In your Data Availability statement, you have not specified where the minimal data set underlying the results described in your manuscript can be found. PLOS defines a study's minimal data set as the underlying data used to reach the conclusions drawn in the manuscript and any additional data required to replicate the reported study findings in their entirety. All PLOS journals require that the minimal data set be made fully available. For more information about our data policy, please see http://journals.plos.org/plosone/s/data-availability.

Response: The data used for the study include sensitive information about the Opioid Substitution Therapy attendees in the ‘Indian Council of Medical Research- Rajendra Memorial Research Institute of Medical Sciences, Patna, India’ (ICMR-RMRIMS). Despite deidentification of the data, markers can be picked up from the dataset through which the identity of the participants may be compromised. Therefore, sharing of data publicly will not be appropriate, as this information will put the participants at risk of being identified and stigmatized. Additionally, the institutional ethics committee of the ICMR-RMRIMS doesn’t allow the publicly sharing of data for publication purpose. However, there is a provision for the bonafide researchers to access the de-identified data after seeking prior approvals from the Principal Investigator & the Director of the ICMR-RMRIMS (Dr. Krishna Pandey Email: drkrishnpandey@yahoo.com; 091-0612-2631565) and the Member secretary of the ethics committee ICMR-RMRIMS, Patna, India (Dr. VNR Das, Email ID: drvnrdas@yahoo.com). 

• Please amend your authorship list in your manuscript file to include author Vidya Nand Rabi Das.

Response: We have mended the authorship list. We have added author Vidya Nand Rabi Das in the manuscript.

• Please include your full ethics statement in the ‘Methods’ section of your manuscript file. In your statement, please include the full name of the IRB or ethics committee who approved or waived your study, as well as whether or not you obtained informed written or verbal consent. If consent was waived for your study, please include this information in your statement as well. 

Response: We have added the ethics statement in the Method section in the main manuscript file. The full name of the ethics committee who have waived the consent: Indian Council of Medical Research- Rajendra Memorial Research Institute of Medical Sciences, Patna, India’ (ICMR-RMRIMS).

REVIEWERS’ COMMENTS AND RESPONSE

Reviewer #1

General comments

Comment 1. This study used medical records from an OST clinic to estimate the HCV prevalence among clients and identify characteristics associated with HCV. The authors use the findings to reiterate their support for increased resources to integrate HCV testing into OST clinics in Bihar specifically, and India generally.

Overall the methods are sound but several steps need clarification. The discussion stretches the beyond the scope of the findings quite a bit. The authors attempt to contextualize their findings within a larger narrative of injection drug use, but offer quite a bit of personal opinion on the matter and use stigmatizing and negative language that is not appropriate.

Response: We appreciate reviewer’s concern for bringing to our notice that the language in the manuscript is stigmatizing and negative towards people with Opioid Use Disorder. We have made appropriate changes in the manuscript to convey a specific message with neutral language. 

Comments 2: The use of the terms ‘addicts’ and ‘addicted’ are outdated and not appropriate the journal’s audience. Individuals with an opioid use disorder is one alternative.

Response: We agree with the reviewer and appreciate for bringing this to our notice. We have removed these terms and used the suggested term Individual with Opioid Use Disorders consistently throughout the manuscript.

Comments 3: “Smack” is not an appropriate term for heroin. Although this is a common street name, readers may interpret it differently.

Response: We have used the correct term impure form of Heroin in place of the term ‘Smack’ throughout the manuscript.

Comment 4: There are substantial typos and grammatical errors throughout; this paper cannot be typeset and published as it is.

Response: We apologise for the English typos and grammatical errors. We have made all the attempts to improve the English language and grammatical errors to improve the flow of the manuscript.

Specific comments for ‘Highlights’ section

Comment 5: “Very high” may be misleading, as this study identified a lower HCV prevalence than was identified in the India-based meta-analysis the authors cited.

Response: We have removed the term ‘Very high’ from the specific statement.

Comment 6: “Effective and evidence-based care can be delivered…” is beyond the scope of this study’s findings. While this is the final recommendation the authors provide, this statement as a highlight should be tempered to align with the study findings

Response: We agree with the reviewer and we have moderated the statement “Effective and evidence-based care can be delivered” to make it more align with the study findings.

Specific comments for ‘Introduction’ section

Comment 7: Lakh should probably be written as thousand for non-Indian readers.

Response: We have removed the term ‘Lakh’ and replaced it with ‘hundred thousand or million’ 

Comment 8: There is a claim that increasing opioid use is attributable to alcohol prohibition. This needs further explanation and a citation.

Response: The news reports in Bihar are constantly reporting that the de-addiction centres in Bihar have been receiving an increased number of cases of substance abuse, ranging from cannabis, inhalants, and sedatives to opioids. This claim is further supported by the specific activities of the state government such as capacity building of Medical Officers of the state by the National Institute of Mental Health and Neurosciences (NIMHANS), Bengaluru to manage the sudden increase in the number of drug abuse cases. 

We have provided the appropriate explanation and citation for our claim.

Comment 9: The government is planning to distribute DAA starting in 2022. Is there more recent data available? Is there any information in the dataset about DAA receipt? This would definitely be a confounder and if not available should be included as a limitation.

Response: The data regarding the DAA distribution through OST or any other center is not available. We agree that receipt of DAA through OST could be a potential confounder and could actually be a reason for the increased enrolment of HCV patients in OST (potentially inflating the HCV seropositive burden among OST attendees). As suggested by the reviewer, we have included this in the ‘limitation’ section.

Specific comments for ‘Methods’ Section

Comment 10: Justify the time frame selected for the medical record extraction. Were previous years or more recent years available, and if so, why were 2014-2020 selected? Clarification on the final sample size is needed. Who was excluded? 25% were lost to follow up – how does this relate to the final analytic sample of n=213?

Response: We have retracted the medical records for the OST attendees till December 2022 and presented the new results based on the data from 2014 to 2022 (N= 268). All OST attendees who had undergone HCV antibody testing (at the time of enrolment) were included for the current analyses. The new results are presented in the updated manuscripts.

Comment 11: The authors state that they did not adjust for drug type because there is no literature supporting its inclusion. The authors didn’t include justification from the literature to include the other variables, so it seems inconsistent. To that end, there certainly is literature around HCV and treatment (buprenorphine) and drug use (heroin and cannabis) which are drug types included in the data. In addition, alcohol has is strongly associated with HCV but is not included. Further explanation and justification for these analytic variable decisions is needed.

Response: We have the removed the statement “we did not adjust for drug type because there is no literature supporting its inclusion” to avoid inconsistencies. 

We had the information for BPN, Pentazocine, Chlorpheniramine (Avil), Heroin, Cannabis, heroin….. We observed that the use of these drugs was not exclusive and one person was consuming more than one drug (using cocktails) at the time of enrolment. Therefore, we created a variable (‘Multiple drug use’) to account for use of more than one drug used it in the regression analyses. We agree with reviewer that the use of alcohol is strongly associated with HCV. However, we don’t have the information about Alcohol use in the existing medical records of study participants. 

Specific comments for ‘Results’ section

Comment 12: The authors make several statements that are not supported by their findings, specifically regarding p-values. For example, stating that clients with HCV were older, but this is not supported by the proportions in Table 1 or by the p-values. This study identified no difference in HCV status by age.

Response: In the revised analyses we have modified the statements.

Comments 13: There needs to be a full description of all the variables included in the study and why. For example, Avil and Penta were not mentioned in the methods section but showed up in the results tables. Why are antihistamines included in the analysis?

Response: We appreciate reviewer for pointing this out. In the method section, under the heading of exposure variables we have mentioned about extracting information related to socio-demographic factors (including age, sex, education level, marital status, and occupation) and drug history (including number of drugs used, type of drug administration, and frequency of drug administration and history of injection sharing). During analyses, we consider all the drugs reported by the patient. In India, individuals with opioid use disorder generally consume opioids along with chlorpheniramine. 

Comments 14: The results include that a higher number of deaths were observed among HCV clients. Death is not defined in the methods section. How was death defined? Death for any reason during the study period? In this case, death would be conceptualized as an outcome, as HCV infection would necessarily precede mortality. Either explain this sub-analysis and the justification for it better, or remove it.

Response: We have removed the parameter ‘patient status’ consisting of on drugs, lost to follow-up, migrated etc. from the analyses. 

Comment 15: The patient status variable in Table 1 is confusing and was not described in the methods section. The term “on drugs” is ambiguous. Also, the inclusion of those who were lost to follow or migrated is difficult to follow. As a cross-sectional study there was no follow up. Are these categories mutually exclusive? Regardless of how you define ‘on drugs’, can a client be on drugs and migrate to another facility? This variable needs to be described in detail in the methods.

Response: We have only considered the cross-sectional component (i.e. data collection at the time of patient enrolment) and removed the follow-up component such as patient’s status (consisting of on drugs, lost to follow-up, migrated etc. from the analyses). 

Comment 16: The resolution of the figure images is very poor and not fit for publication. Provide these to the journal as vector graphics.

Response: We have uploaded a better quality figure images.

Comment 17: The trend for age is not significant, but stated using language that suggests a trend was observed. The same for years of injection use. It is also unclear which model these trends were derived from, which should be clarified in the methods.

Response: We have modified the statement describing the trend of HCV seropositive status by age categories. We assigned mid-point to each specified age-group categories (such as 19 for the age group category 18 to 20, 23 for the age group category 21 to 25; 28 for the age group category 26 to 30; 33 for the age group category 31 to 35; 38 for the age group category 36 to 40; and 50 for the age group category 41 to 59); and treated it as a continuous variable to assess the linear trend across increasing age-categories using robust Poisson regression model.

Comment 18: Injection was significant in the full model, but it is not stated clearly where the estimates for the figures were derived from.

Response: The estimates for the figures was calculated based on the univariable robust Poisson regression model. We assigned specific value to each years of injection use categories and treated it as a continuous variable to assess the linear trend across increasing years of injection use using a robust Poisson regression model.

Comment 19: The decision to conduct two separate models – restricted and full – needs more rationale and justification as to what one model tells us that the other cannot. But, it seems as if the authors only present the results from the full model anyway, so the purpose of the restricted is unknown. The variable selection criteria of p<0.10 from the univariable models needs a citation. 

Response: We appreciate reviewer for pointing out the redundancy in the analyses. We have removed the restricted model from the analyses

Specific comments for ‘Discussion’ Section - Burden paragraph

Comment 20: The authors state that patients in OST don’t represent true population of IDUs – why? This needs to be explained further and supported with citations.

Response: All the participants enrolled in the OST center in ICMR-RMRIMS were males. Evidence suggest that at a given point in time only 5% of opioid drug users receives OST mainly delivered by community-based services. 

Comment 21: The authors state that OST and non-OST populations differ because the former is more responsible. This is not supported in the literature and is highly judgmental and stigmatizing and language that perpetuates clinical treatment disparities. This is an opinion and not appropriate for publication.

Response: We appreciate the reviewer for highlighting our mistake. We have modified the language in revised manuscript and made it more neutral.

Comment 22: The authors state that they can confidently say they underestimated HCV – why? How? Overestimating is also plausible, in that people with more severe disorders and higher HCV risk are the ones who get treatment. How the authors are conceptualizing underestimation, and how they use the epidemiologic and drug use literature to back it up, are needed.

Response: We agree with the reviewer that the HCV seropositivity estimate could have been underestimated or overestimated. We have modified the statement in the revised manuscript.

Comment 23: The statement “IDU live alone in poverty”, and several other similar statements in this paragraph, is not supported by the literature, which is likely the reason the authors offer no citations. Living alone and poverty are certainly determinants of drug use, but these characteristics do not universally describe the IDU population.

Response: We agree with the reviewer and have modified the statement in the revised manuscript

Comment 24: History of crime is also not cited – again, this is stigmatizing language and projects a judgement from the authors. Poverty, exposure to the criminal justice system, and the resulting stress all drive drug use. The relationship between crime and drug is complex and requires considerable nuance, tact, and literature review to discuss appropriately

Response: We appreciate reviewer for highlighting our mistake. We have modified the statements in discussion section to avoid stigmatizing language.

Specific comments for ‘Discussion’ - Determinants paragraph

Comment 25: The claim that higher HCV indicative of worsening HCV burden is misleading and not supported by the study findings. The study found a prevalence lower than the 44.7% cited by the authors (Goel et al.). In addition, as stated earlier, the time trend was not significant. A disproportionately high and growing HCV burden is the crux of the paper and in the title, but doesn’t align with the study findings.

Response: We have modified the specified section in the revised manuscript.

Specific comments for ‘Discussion’ - Public health paragraph

Comment 26. The authors suggest that OST is an ideal opportunity to screen for HCV. This is a great point and much of the discussion would be substantially strengthened if this were described in greater detail. People who use and inject drugs are notoriously hard to reach in communities, and there are few places where effective screening may reach a large portion of the population. OSTs are one of the few places that provide that opportunity.

Response: We appreciate reviewer for his/her suggestion. We have made the appropriate changes in the discussion section- public health implication paragraph.

Response to Reviewer #2: 

Comment 1: Two of the highlights are not supported by the data presented in the manuscript: 

- The opioid burden is rising in Bihar -- Evidence to support this is anecdotal and insufficiently detailed to include this point as a highlight. 

- Effective and evidence-based care can be delivered by integrating HCV care into OST-center – nothing is presented about care delivery; instead, the manuscript suggests that the authors feel that there an opportunity to do. This argument is presented in the Discussion as is appropriate, but it is not a highlight of the study. 

Response: We agree with the reviewer. We have modified the stated statements.

Comment 2. The authors use the term “injection drug user(s)” and the abbreviation IDU. It would be better to use the term “people who use drugs” and the abbreviation “PWID”, which describe the medical condition of these individuals rather than define them as human beings. PWID should be used throughout the manuscript. Also, the term “addict” is pejorative and is to be avoided. The individuals in the study have been diagnosed with and are being treated for opioid use disorder (OUD) and should be described as such. 

Response: We have replaced the term injection drug user(s) with the term people who use drugs (PWID) and used it throughout the manuscript to maintain the consistency. Also, we have deleted the term ‘addict’ from the manuscript. In the revised manuscript, the participants are described as ‘individuals with opioid use disorder.’

Comment 3. The authors write, “Bihar, one of the eastern states in India, has the least prevalence of opioid use (0.1%) compared to the national average of 0.7% [8]. Citation is about cardiovascular disease, not opioid use. Nothing in the cited article is relevant to this point the authors wish to make. Either identify a correct reference or delete.

Response: We appreciate reviewer for pointing this out. Mistakenly, we have inserted an incorrect citation. In the modified manuscript we have inserted the correct citation. The correct reference is “Magnitude of Substance Use in India, 2019”

Comment 4: The authors report that OST is initiated by a medical officer if an individual meets a set of criteria, one of which includes the individuals being “[c]urrent IDUs”. Does this mean that people with OUD but who are injecting drugs are excluded from treatment? If those not injecting are eligible for treatment, then the authors need to make clear that their sample is restricted to those patients who report a history of injection. Also, the authors report the age for eligibility as >18; does this mean that those 18 years of age are excluded. This has implications for how subsequent data are reported. In Table 1 the youngest age category is ≤20; in Figure 1 it is <20. Which is correct? And if there is a lower age cut off (either 18 or 19), then category should best be reported as 1x-20 or 1x-19, and if those 18 years of age are excluded, then the youngest category would be either “19” or “19-20”. Furthermore, the authors need to comment on the fact that nearly 20% of their sample fall into a category that covers a very few number of ages: some or all of those aged 18-20. It is quite unusual for a treatment-based sample to have such a high percentage of individuals who are so young. 

Response: In the OST center in ICMR-RMRIMS, Patna, India, the medical officer enrol only OUDs who are currently injecting drugs. We agree that the sample for this study is restricted to those patients who report a history of injection. The age of eligibility is 18 years and older. We have realized the mistake we committed during data analyses. We have made appropriate changes in the revised manuscript. The youngest category is 18 to 20 years. As suggested by the reviewer, we have included a statement describing nearly 12.7% of the total participants fall into an age category of 18-20 years.

Comment 5: In the Methods, the authors report that age was expressed as means and standard deviation (SD). This is not strictly speaking correct. This is only true for the presentation of the data in the top row of Table 1 and (I believe) in Table 2. It is presented subsequently in Table 1 and in Figure 1 as a categorical variable. This needs clarification. 

Response: In the statistical analyses section, we have mentioned that age was treated as continuous as well as categorical variable to assess its association with HCV status. 

Comment 6: The authors describe their study design as cross-sectional, but they are including longitudinal data beyond the baseline when they introduce the results on mortality in their sample. The authors can either revise the study methods to mention that follow-up mortality data were obtained or they can delete this sentence since the difference in mortality as a function of HCV seropositivity was not found to be significant. 

Response: We appreciate the reviewer for pointing out this mistake. In the revised manuscript, we have modified the method section. We have only considered the cross-sectional component (i.e. data collection at the time of patient enrolment) and removed the follow-up component such as patient’s status (consisting of on drugs, lost to follow-up, migrated etc. from the analyses). 

Comment 7. In the text, the authors report that “There was a rising prevalence of hepatitis C with age (p- trend= 0.811).” But this trend does not appear to be significant and since the regression is linear, the authors should report the r (or r2) statistic as well as the p-value. Also, regarding Figure 1, the curve depicting HCV prevalence should not begin at the origin since the first real data point is for individuals aged <20, but it could be for those aged 18-19, 18-20, or just those aged 19. Please be sure that reporting of this youngest age group is consistent in the Methods, Results, Table 1, and Figure 2. 

Response: We have corrected the analyses and make appropriate changes in the modified manuscript. To assess the increasing trend of HCV seropositive status by increasing age, we used robust Poisson regression model and not the linear regression. Therefore, we cannot report the r2 statistics.

Comment 8. In the Discussion, the authors state, “We also found that younger patients (<35 years of age)...had a higher burden of HCV than their counterparts.” The statistical analysis in the model presented in Table 2 does not support this conclusion; instead, age does not appear to be statistically associated with prevalence. In this part of the discussion, the authors could comment that the small sample size might have had insufficient power to identify trends associated with age. 

Response: As suggested by the reviewer, we have modified the statement.

Comment 9. In the Discussion, the authors opine that, “OST attendees or IDUs seeking services from any de-addition center differs from their counterparts; Such as they are more responsible, more educated, have more family and friend support and higher socio-economic status, and have less needle sharing behaviour etc.” No data are presented to support the suggested differences. The authors’ speculations may represent their prejudices more than they fairly describe differences between OTP patients and PWID not in treatment. If the authors want to keep this sentence, the directionality of differences between patients and non-patients should be directionless. For example, they could state that the two groups COULD differ in education attainment, social support, and behavioural risks. 

Response: We have inserted appropriate citations to support our claim. Also, as suggested by the reviewer we have modified the statement and made it directionless.

Comment 10. In the Discussion, the authors comment that, “...we have also observed a very high HCV sero-prevalence among HIV-positive patients: 58.3% (26.7% - 84.3%). Such finding is expected as HIV significantly alters the pathological course of HCV [11, 12]. HCV patients co- infected with HIV show increased viral load, hepatic decompensation and mortality from the complications of end-stage hepatic diseases compared to HCV mono-infected individuals [13-16].” All of this is irrelevant, especially since the difference in HIV status as a function of HCV status was not significant (p=0.115 as reported in the text and in Table 1). The study is not assessing the course of HCV-related liver disease. The diagnostic test does not even reveal if a patient who tested seropositive is actively infected or has cleared the virus. Therefore, there is no need for any discussion of disease progression or mortality. It is enough to state that the finding of higher HCV prevalence in those co-infected with HIV is consistent with global data and that HCV infection frequently precedes HIV infection because HCV is more infectious.

Response: As suggested by the reviewer, we have made appropriate changes in the discussion section.

Comment 12. As noted earlier, the curves presenting the prevalence of HCV and confidence intervals in Figures 1 and 2 should not begin at the origin but at the first data point. 

Response: We appreciate reviewer for pointing this out. We have made appropriate changes in both Figures and uploaded the corrected one.

Comment 13. Below are the typographical, grammatical, and semantical errors that need to be addressed, again listed by the appearance in the pdf of the manuscript.

Response: We appreciate the reviewer for pointing this out. We have made appropriate changes in the modified manuscript.

Background and Rationale 

Comment 13.1. In the first paragraph, the text reads, “Thus at a given point in time, ~55 lakh patients are HCV seropositive and over the period of next 20 to 30 years ~15 lakh of these will develop chronic hepatic illnesses...” What is “lakh”? I assume it is the standard epidemiological denominator of 100,000. It would be better to use the numerical formulation since lakh seems to be specific to Indian English (according to Oxford Languages). 

Response: We have replaced the work lakh with the standard epidemiological denominator of 100,000.

Comment 13.2. The authors write, “Opioid substitution therapy (OST) canters/clinics have been developed to support opioid addicts (IDUs) to quit...” It would be more consistent with currently agreed upon terminology to revise this to read, “Opioid substitution therapy (OST) centers/clinics have been developed over the course of the last three decades to treat individuals with opioid use disorder (OUD) and have now escalated...” 

Response: We have modified the statement as per reviewer’s suggestion.

Methods

Comment 13.3. The authors write, “The OST clinic started at the RMRIMS, Patna since January 2014 as a part of NACO (funded by Ministry of Health & Family Welfare, India) to reduce the transmission of HIV among IDUs.” This sentence should read, “The OST clinic was started at the RMRIMS, Patna in January 2014 as a part of NACO’s mission (funded by Ministry of Health & Family Welfare, India) to reduce the transmission of HIV among PWID.” 

Response: We have modified the statement as per reviewer’s suggestion.

Results

Comment 13.4. There is an opening parenthesis missing in the 2nd sentence of the first paragraph.

Response: We have modified the statement as per reviewer’s suggestion.

Comment 13.5. The authors write, “Participants with HCV positive status (compared to HCV negative status) were older (p-value= 0.732), unmarried or separated or divorced (p-value =0.122)...” Neither of these differences were significant and the text should reflect point; the authors can choose not to report these comparisons (as the data appear in Table 1) or they must state the differences were not statistically significant in the text. 

Response: We have modified the statements as per reviewer’s suggestion.

Comment 13.6. Please define Avil and Penta using generic drug names for the international audience here and in Table 1. Also, please use a non-slang term for “smack” in Table 1. Does smack refer to heroin? 

Response: We have defined the term Avil, smack and Penta in the modified manuscript.

Comment 13.7. The authors write, “Participants with HCV positive status also had higher burden of HEV (p- value= 0.025) and HIV (p-value= 0.115).” This sentence should be corrected to read, “Participants with HCV positive status also had a higher burden of HEV (p-value= 0.025) but not of HIV (p-value= 0.115).” 

Response: We have modified the statement as per reviewer’s suggestion.

Comment 13.8. In presenting the data in Table 1, since statistical testing is a comparison by HCV status, it would make more sense to have the totals column to the left, then two groups by HCV status before reporting the results of the statistical test. 

Response: We have modified the table 1 as per reviewer’s suggestion.

Comment 13.9. In Table 1, the row heading should read “Total number of drugs use, median (IQR)”.

Response: We have modified the table 1 as per reviewer’s suggestion

Comment 13.10. The authors write, “There was a rising prevalence of hepatitis C with number of years of injection use (p- trend= 0.256).” The trend is not statistically significant. The text should explicitly mention that the increase in prevalence was not significantly associated with number of years injecting. Discussion 

Response: We have modified the statement as per reviewer’s suggestion.

Comment 13.11. The parenthetical phrase “(who are addicted to injectable drugs)” should be deleted. 

Response: We have made appropriate changes as per reviewer’s suggestion.

Comment 13.12. The authors write, “The observed HCV prevalence among OST is little bit lesser than the prevalence reported in the meta-analyses by Goel et al...” This should be revised to read, “The observed HCV prevalence 34.7% (95% CI: 28.3%-41.5%) among OST is somewhat less than the prevalence reported in the meta-analyses by Goel et al...” 

Response: We have modified the statement as per reviewer’s suggestion.

Comment 13.13. In the sentence that begins: Nevertheless, our finding suggests that every third patient...” “demand” should be revised to “demands”. 

Response: We have modified the statement as per reviewer’s suggestion.

Comment 13.14. Substantial editing is needed for the sentences that read:

Several clinical, social and behavioural factors drive the higher prevalence of HCV among IDUs. These are: many IDUs are younger, homeless and live alone in poverty; have a higher mental health problems; do not seek or cannot receive quality healthcare without assistance. Additionally, many have history of crime, so do not access any social services for fear of being discovered and returning to prison. Appropriate context-specific interventions targeting this high risk population should be explored.

I suggest:

Several clinical, social and behavioural factors drive the higher prevalence of HCV among IDUs. Many PWID are young [no need to use the comparative], homeless and live alone in poverty. They may have a more mental health problems, and, as a consequence do not seek or cannot receive quality healthcare without assistance. Many may have history of crime, and so do not access social services for fear of being discovered and returning to prison. Appropriate context-specific interventions targeting this high-risk population should be explored. 

Response: We have modified the statement as per reviewer’s suggestion.

Comment 13.15. The authors write, “The higher HCV sero-prevalence in our study suggests a worsening of HCV burden among IDUs very similar to studies conducted in other settings.” This sentence does not make sense to me. The HCV prevalence is consistent other studies of PWID; it is not higher. No evidence is presented that the burden is increasing/worsening over time. I suggest either completely revising this sentence or deleting it. 

Response: We have removed the specified statement as per reviewer’s suggestion.

Comment 13.16. The authors write, “Similarly, HCV sero-prevalence was...higher among HIV positive patients. The difference as reported in Table 1 was not significant, so this point should be deleted from the sentence. 

Response: We have modified the statement as per reviewer’s suggestion.

Comment 13.17. In point (e) in the paragraph on public health significance, “HIV care across car continuum” should be revised to read “HIV care across the continuum”. 

Response: We have modified the statement as per reviewer’s suggestion.

Comment 13.18. The References need to be carefully reviewed for format, capitalization, accuracy. Any citation of website or non-peer reviewed document that is not accessible with a DOI number should have a URL attached and a date that the URL was last accessed. For example, citation 5, “[5] Centers for disease control and prevention. Hepatitis c faqs for the public. 2022”. This should be revised to state the publishing organization the US centers for disease control and prevention. (I understand that the British convention is to capitalize only the first word in an organization or official title, but for ease of understanding, I think the letter identifying a specific hepatitis visus should always be capitalized (or is it capitalised?)) The title should read Hepatitis C FAQs, and URL and last access date needs to be added. 

Response: We have modified the references as per reviewer’s suggestion.

---

## [Decision Letter · Decision Letter 1]

18 May 2023

PONE-D-22-32802R1The burden of Hepatitis-C virus infection among patients enrolled at the opioid substitution therapy center in Bihar: A cross-sectional studyPLOS ONE

Dear Dr. Pandey,

Thank you for submitting your manuscript to PLOS ONE. After careful consideration, we feel that it has merit but does not fully meet PLOS ONE’s publication criteria as it currently stands. Therefore, we invite you to submit a revised version of the manuscript that addresses the points raised during the review process.

We look forward to receiving your revised manuscript.

Kind regards,

Nickolas Zaller

Academic Editor

PLOS ONE

Journal Requirements:

Reviewers' comments:

Reviewer's Responses to Questions

**Comments to the Author**

1. If the authors have adequately addressed your comments raised in a previous round of review and you feel that this manuscript is now acceptable for publication, you may indicate that here to bypass the “Comments to the Author” section, enter your conflict of interest statement in the “Confidential to Editor” section, and submit your "Accept" recommendation.

Reviewer #1: All comments have been addressed

Reviewer #2: (No Response)

2. Is the manuscript technically sound, and do the data support the conclusions?

Reviewer #1: Yes

Reviewer #2: Partly

3. Has the statistical analysis been performed appropriately and rigorously? 

Reviewer #1: Yes

Reviewer #2: Yes

4. Have the authors made all data underlying the findings in their manuscript fully available?

Reviewer #1: Yes

Reviewer #2: No

5. Is the manuscript presented in an intelligible fashion and written in standard English?

Reviewer #1: Yes

Reviewer #2: No

6. Review Comments to the Author

Reviewer #1: This revised version is substantially improved, the authors have responded to all of my original concerns. I do not have any further comments or critique.

Reviewer #2: Items to attend to in Bihar HCV prev mss

Introduction

1) I recommend changing the title to “Hepatitis C virus seroprevalence…” SInce about 30% of seropositive individual clear the virus, testing for antibodies does not accurately reflect the burden of infection but rather just evidence of past infection that could have become chronic.

2) Intro paragraph 1, comment on prompt treatment for HCV: No, prompt treatment is not necessary as time between infection and life-threatening manifestations of end-stage disease in measured in decades.

3) Paragraph 2 “Health care related procedures, injection drug use, and blood transfusions contribute to the spread of the HCV virus” delete “virus”

4) Revise “HCV transmission has reduced significantly through blood transfusion” to read, “HCV transmission through blood transfusion has been significantly reduced….”

5) Paragraph 3 text that reads “Traditionally, it has been seen that ~15% PWID have HIV/AIDS…” should be revised to read “Traditionally, it has been seen that ~15% of PWID has HIV/AIDS…”

6) Paragraph 4, last sentence currently reads, “…it is important to have a good quality epidemiological data with reasonable accuracy.” Revise to read, ““It is important to have reasonably accurate epidemiological data.”

Material and Methods

1) Study population, 1st paragraph – text reads “Whenever a patient visits OST canter (to seek services)”, fix typo “canter”.

2) Study population, 1st paragraph – text reads “All patients at baseline also undergo for HIV, HBV antigen, and HCV antibody tests,” Revise the read “All patients at baseline also undergo testing to detect HIV, HBV antigen, and HCV antibody.

Results

1) First paragraph – In the sentence, “The HCV seropositivity prevalence was 28% [95% CI: 22.7% - 33.8%)].” Delete the bracket at the end.

2) Generic and street drug names should not be capitalized in the text; okay to capitalize a row name in the tables.

3) Age is analyzed in two ways – as means via unpaired t-test and as a categorical variable. In the second paragraph of the Results, it appears that the analysis of age used the t-test. It would clarify things if this were indicated by a footnote in Table1.

4) Third paragraph – Although the test for trend is significant, there really is not rising prevalence. The sentence that read “There was a rising prevalence of HCV seropositivity with age (p-trend= 0.025).” should be deleted. The next sentence better describes the situation. Prevalence was lower in the two youngest age groups compared to the four older ones, which were quite similar.

5) Third paragraph – Fix typo in next sentence to read: Patients aged 25 years or less had..”

6) In the fourth paragraph, the 2nd and 3rd sentences are inconsistent with each other. Prevalence is higher in each group with more years of injecting compared to the group with fewer years — a monotonic increase in prevalence. The sentence about a sharp increase among those injecting >2 years suggests that thereafter prevalence did not increase much after that first increase after 2 years injecting, but this is not what that the data show. Delete the sentence about the sharp increase.

7) In the fifth paragraph, fix typo to read, “Patients with education up to higher…”

8) The last sentence in the fifth paragraph about the association of longer time injecting and HCV seropositivity should be moved to the previous paragraph discussing the association between years of injection and HCV seroprevalence

Discussion

1) First paragraph – fifth sentence should be revised to read “We also found that those unemployed, less educated, and who were using injection drugs for longer period of time had a higher HCV seropositivity than their counterparts.”

2) Second paragraph – In the second sentence delete the mention of the confidence interval and the sentence can be rephrased to read “The observed HCV prevalence of 28% among OST patients in Bihar was significantly less than the 44.7% prevalence reported by Goel et al. in their systematic analysis of HCV in PWID across India.”

3) In suggesting reason for the lower prevalence, you should note that geographic differences may also play a role.

4) Second paragraph – Revise sentence to read “Nevertheless, our findings suggest that approximately 30% of patients….”

5) Second paragraph, last sentence – revise to read “Unfortunately, the available clinical records for the OST attendees in ICMR-RMRIMS do not provide any information about adverse exposures, prohibiting comparison with other studies.”

6) Public health significance – The first sentence is one very long run-on with five distinct parts. Each should be its own sentence.

7) Strengths and limitations -- A major limitation is the very small sample size. The clinic enrolled, on average, fewer than 30 patients every year. The barriers to entry — daily or near daily visits, active injection, failure of non-medication treatment for OUD — made for not only a small but potentially very biased sample.

Conclusion

1) Delete “a” from the 3rd sentence “…where a comprehensive HCV care…”

Figures

1) Both figures need to be revised. The current figures are not the correct way to present the data on HCV prevalence and age or years of injection, and the presentation actually complicates interpretation of the data. Since age and years of injection are presented as categorical variables, the HCV data should be represented by a bar with the confidence interval stretching above and below the bar. There is no way that prevalence and 95% CI should be shown as curves even if data were analyzed using Poisson regression. There is no need to present the data on the %age of study participants in each category (age and years of injection) as a bar; this information could be placed in text box above bar.

7. PLOS authors have the option to publish the peer review history of their article (what does this mean?). If published, this will include your full peer review and any attached files.

Reviewer #1: No

Reviewer #2: No

---

## [Author Response · Author response to Decision Letter 1]

24 May 2023

RESPONSE TO EDITOR’S AND REVIEWERS’ COMMENTS

Editor’s comments

• A rebuttal letter that responds to each point raised by the academic editor and reviewer(s). You should upload this letter as a separate file labelled 'Response to Reviewers'

Response: Appropriate rebuttal letter named 'Response to Reviewers' has been drafted and submitted at the designated PlosOne submission portal.

• A marked-up copy of your manuscript that highlights changes made to the original version. You should upload this as a separate file labelled 'Revised Manuscript with Track Changes'.

Response: A revised document with all the suggested modifications highlighted in Yellow has been drafted, labelled as 'Revised Manuscript with Track Changes, and uploaded at the PlosOne submission portal.

• An unmarked version of your revised paper without tracked changes. You should upload this as a separate file labelled 'Manuscript'.

Response: A revised document with all the suggested modifications has been drafted, labelled as ‘Manuscript’, and uploaded at the PlosOne submission portal.

Journal Requirements

Response: We have reviewed the reference list. It is complete and correct. We have not cited any paper that has been retracted. We have added two new references (Ref 5 and Ref 11) in the reference list and highlighted in yellow. 

Reviewers’ comments and response

Reviewer #2

Specific comments for ‘Introduction’ section

Comment 1. I recommend changing the title to “Hepatitis C virus seroprevalence…” SInce about 30% of seropositive individual clear the virus, testing for antibodies does not accurately reflect the burden of infection but rather just evidence of past infection that could have become chronic.

Response: We appreciate reviewer’s suggestion and changed the title to “Hepatitis C virus seroprevalence among patients enrolled at the opioid substitution therapy center in Bihar: A cross-sectional study”

Comment 2. Intro paragraph 1, comment on prompt treatment for HCV: No, prompt treatment is not necessary as time between infection and life-threatening manifestations of end-stage disease in measured in decades.

Response: We believe that the presence of HCV acute infection warrants expedited linkage to care with a specialist engaged in the management of hepatitis patients. The spontaneous clearance of HCV infection generally occurs in patients treated within 6 months of infection and significantly reduces the chances of life-threatening manifestations of HCV infection. (Ref 5. Guss D et al. Diagnosis and Management of Hepatitis C Infection in Primary Care Settings. Journal of general internal medicine 2018.)

Comment 3. Paragraph 2 “Health care related procedures, injection drug use, and blood transfusions contribute to the spread of the HCV virus” delete “virus”

Response: We have deleted the term virus after HCV.

Comment 4. Revise “HCV transmission has reduced significantly through blood transfusion” to read, “HCV transmission through blood transfusion has been significantly reduced….”

Response: We have revised the statement “HCV transmission has reduced significantly through blood transfusion” to “HCV transmission through blood transfusion has been significantly reduced” in the manuscript.

Comment 5. Paragraph 3 text that reads “Traditionally, it has been seen that ~15% PWID have HIV/AIDS…” should be revised to read “Traditionally, it has been seen that ~15% of PWID has HIV/AIDS…”

Response: We have revised the statement “Traditionally, it has been seen that ~15% PWID have HIV/AIDS…” to “Traditionally, it has been seen that ~15% of PWID has HIV/AIDS…” in the main manuscript.

Comment 6. Paragraph 4, last sentence currently reads, “…it is important to have a good quality epidemiological data with reasonable accuracy.” Revise to read, ““It is important to have reasonably accurate epidemiological data.”

Response: We have revised the statement ““…it is important to have a good quality epidemiological data with reasonable accuracy” to “It is important to have reasonably accurate epidemiological data …” in the main manuscript.

Specific comments for ‘Methods’ Section

Comment 1. Study population, 1st paragraph – text reads “Whenever a patient visits OST canter (to seek services)”, fix typo “canter”.

Response: We have fixed the typo “canter”.

Comment 2. Study population, 1st paragraph – text reads “All patients at baseline also undergo for HIV, HBV antigen, and HCV antibody tests,” Revise the read “All patients at baseline also undergo testing to detect HIV, HBV antigen, and HCV antibody

Response: We have revised the statement “All patients at baseline also undergo for HIV, HBV antigen, and HCV antibody tests,” to “All patients at baseline also undergo testing to detect HIV, HBV antigen, and HCV antibody.

Specific comments for ‘Results’ section

Comment 1. First paragraph – In the sentence, “The HCV seropositivity prevalence was 28% [95% CI: 22.7% - 33.8%)].” Delete the bracket at the end.

Response: We have deleted the bracket.

Comment 2. Generic and street drug names should not be capitalized in the text; okay to capitalize a row name in the tables.

Response: As suggested, we have made the appropriate changes in the manuscript.

Comment 3. Age is analyzed in two ways – as means via unpaired t-test and as a categorical variable. In the second paragraph of the Results, it appears that the analysis of age used the t-test. It would clarify things if this were indicated by a footnote in Table1.

Response: As suggested, we have made the appropriate changes in the manuscript. We have added clarifying footnote in Table 1.

Comment 4. Third paragraph – Although the test for trend is significant, there really is not rising prevalence. The sentence that read “There was a rising prevalence of HCV seropositivity with age (p-trend= 0.025).” should be deleted. The next sentence better describes the situation. Prevalence was lower in the two youngest age groups compared to the four older ones, which were quite similar.

Response: We have made appropriate changes in the manuscript.

Comment 5. Third paragraph – Fix typo in next sentence to read: Patients aged 25 years or less had..”

Response: We have made appropriate changes in the manuscript.

Comment 6. In the fourth paragraph, the 2nd and 3rd sentences are inconsistent with each other. Prevalence is higher in each group with more years of injecting compared to the group with fewer years — a monotonic increase in prevalence. The sentence about a sharp increase among those injecting >2 years suggests that thereafter prevalence did not increase much after that first increase after 2 years injecting, but this is not what that the data show. Delete the sentence about the sharp increase.

Response: We have made appropriate changes in the manuscript.

Comment 7. In the fifth paragraph, fix typo to read, “Patients with education up to higher…”

Response: We have made appropriate changes in the manuscript.

Comment 8. The last sentence in the fifth paragraph about the association of longer time injecting and HCV seropositivity should be moved to the previous paragraph discussing the association between years of injection and HCV seroprevalence

Response: Thank you for the suggestion. The findings in the fourth paragraph are describing the unadjusted association between years of injection use and HCV prevalence. Whereas, the fifth paragraph is describing the association between years of injection and HCV seroprevalence after adjusting for the effect of sociodemographic and other clinical variables. 

Specific comments for ‘Discussion’ Section

Comment 1. First paragraph – fifth sentence should be revised to read “We also found that those unemployed, less educated, and who were using injection drugs for longer period of time had a higher HCV seropositivity than their counterparts.”

Response: We have made appropriate changes in the manuscript.

Comment 2. Second paragraph – In the second sentence delete the mention of the confidence interval and the sentence can be rephrased to read “The observed HCV prevalence of 28% among OST patients in Bihar was significantly less than the 44.7% prevalence reported by Goel et al. in their systematic analysis of HCV in PWID across India.”

Response: We have made appropriate changes in the manuscript.

Comment 3. In suggesting reason for the lower prevalence, you should note that geographic differences may also play a role.

Response: As suggested we have added the required information (with appropriate citation) in the discussion section.

Comment 4. Second paragraph – Revise sentence to read “Nevertheless, our findings suggest that approximately 30% of patients….”

Response: We have made appropriate changes in the manuscript.

Comment 5. Second paragraph, last sentence – revise to read “Unfortunately, the available clinical records for the OST attendees in ICMR-RMRIMS do not provide any information about adverse exposures, prohibiting comparison with other studies.”

Response: We have made appropriate changes in the manuscript.

Comment 6. Public health significance – The first sentence is one very long run-on with five distinct parts. Each should be its own sentence.

Response: We have broken down the long sentence into four to five small sentences. 

Comment 7. Strengths and limitations -- A major limitation is the very small sample size. The clinic enrolled, on average, fewer than 30 patients every year. The barriers to entry — daily or near daily visits, active injection, failure of non-medication treatment for OUD — made for not only a small but potentially very biased sample.

Response: We really appreciate reviewer’s suggestion. We have added the limitation of small sample size in the revised manuscript.

Specific comments for ‘Conclusion’ Section

Comment 1. Delete “a” from the 3rd sentence “…where a comprehensive HCV care…”

Response: We have made appropriate changes in the manuscript.

Specific comments for ‘Figures’

Comment 1. Both figures need to be revised. The current figures are not the correct way to present the data on HCV prevalence and age or years of injection, and the presentation actually complicates interpretation of the data. Since age and years of injection are presented as categorical variables, the HCV data should be represented by a bar with the confidence interval stretching above and below the bar. There is no way that prevalence and 95% CI should be shown as curves even if data were analyzed using Poisson regression. There is no need to present the data on the %age of study participants in each category (age and years of injection) as a bar; this information could be placed in text box above bar.

Response: As per reviewer’s suggestion, we have modified figures (1 & 2) and put the proportion of participants for ‘age group’ categories as well as ‘years of injection use’ categories in footnote for Figure 1 and 2.

---

## [Editor Report · Decision Letter 2]

4 Jun 2023

Hepatitis C virus seroprevalence among patients enrolled at the opioid substitution therapy center in Bihar: A cross-sectional study

PONE-D-22-32802R2

Dear Dr. Pandey,

We’re pleased to inform you that your manuscript has been judged scientifically suitable for publication and will be formally accepted for publication once it meets all outstanding technical requirements.

Kind regards,

Nickolas Zaller

Academic Editor

PLOS ONE
---

## [Editor Report · Acceptance letter]

8 Jun 2023

PONE-D-22-32802R2 

Hepatitis C virus seroprevalence among patients enrolled at the opioid substitution therapy center in Bihar: A cross-sectional study 

Dear Dr. Pandey:

I'm pleased to inform you that your manuscript has been deemed suitable for publication in PLOS ONE. Congratulations! Your manuscript is now with our production department. 

Kind regards, 

on behalf of

Dr. Nickolas Zaller 

Academic Editor

PLOS ONE